# Pore partition in two-dimensional covalent organic frameworks

Xiaoyi Xu [1], Xinyu Wu[1], Kai Xu[1], Hong Xu [2], Hongzheng Chen [1] & Ning Huang [1] ✉

Covalent organic frameworks (COFs) have emerged as a kind of crystalline polymeric materials with high compositional and geometric tunability. Most COFs are currently designed and synthesized as mesoporous (2–50 nm) and microporous (1–2 nm) materials, while the development of ultramicroporous (<1 nm) COFs remains a daunting challenge. Here, we develop a pore partition strategy into COF chemistry, which allows for the segmentation of a mesopore into multiple uniform ultramicroporous domains. The pore partition is implemented by inserting an additional rigid building block with suitable symmetries and dimensions into a prebuilt parent framework, leading to the partitioning of one mesopore into six ultramicropores. The resulting frame-work features a wedge-shaped pore with a diameter down to 6.5 Å, which constitutes the smallest pore among COFs. The wedgy and ultramicroporous one-dimensional channels enable the COF to be highly efficient for the separation of five hexane isomers based on the sieving effect. The obtained average research octane number (RON) values of those isomer blends reach up to 99, which is among the highest records for zeolites and other porous materials. Therefore, this strategy constitutes an important step in the pore functional exploitation of COFs to implement pre-designed compositions, components, and functions.

Covalent organic frameworks (COFs), as an attractive family of crystalline porous polymers with structural flexibility, high porosity, and functional versatility[1–5], have caught extensive concern in the fields of gas adsorption[6–9], heterogeneous catalysis[10–13], optoelectronics[14–16], sensing[17–19], and energy storage[20–22]. Reticular chemistry allows chemists to implement sophisticated designs and precise control over their structures and functionalities[23,24]. Despite hundreds of COFs developed in past decades, most of them are mesoporous with pore sizes ranging from 2 to 5 nm[25,26]. Owing to the scarcity of undersized building blocks and thermodynamic instability, the number of microporous (<2 nm) COFs is relatively limited, especially ultramicroporous ones with pore sizes less than 1 nm. It is well known that ultramicropores in an angstrom scale can selectively enhance the interaction between pore walls and certain gas molecules. Ultramicroporous materials have received serious concerns in gas separation and purification, as they can implement the molecule-scale separation of various gases, such as carbon dioxide/nitrogen[27–29], propyne/propylene[30–32], and krypton/xenon[33–35]. Therefore, it's crucial to develop a feasible synthetic strategy to construct pore architectures that allow for efficient utilization of pore space via pore partitioning of large channels so as to mitigate the demands on ultramicroporous materials in terms of gas separation.

It represents a more advanced stage of COFs to introduce additional functional units into their original skeletons, realizing fine control over their properties for specific aims. In this respect, pore surface engineering provides a useful approach for post modification of COF

[1]State Key Laboratory of Silicon and Advanced Semiconductor Materials, International Research Center for X Polymers, Department of Polymer Science and Engineering, Zhejiang University, 310027 Hangzhou, China. [2]Institute of Nuclear and New Energy Technology, Tsinghua University, 100084 Beijing, China. ✉e-mail: nhuang@zju.edu.cn

pore walls with various functional units[36–38]. The functionalization can be implemented by the employment of azide, alkyne, alkene, or phenol appended building blocks for the synthesis of COFs, thereby implementing precise control over the content of these functional groups. These anchored groups can further undergo quantitative reactions to create pore surfaces with conceivable functionalities. Although surface engineering strategy can serve as an efficient tool to realize the regulation over internal physical and chemical environments of one-dimensional (1D) channels, it proves feeble for the transformation of pore sizes and pore shapes[39,40], probably due to the flexible and adaptive nature of appended functional groups. From the perspective of topological diagrams, it's viable to insert additional rigid building blocks with suitable symmetries and dimensions to partition one mesopore into several micropores or ultramicropores (Fig. 1a). To put this hypothesis into practice, we anchored a secondary active site at the linkers and then run hooks from the center of framework to these reactive groups, consequently triggering the second reaction and partitioning a hexagonal pore into multiple uniform domains (Fig. 1b). Such a pore partition strategy offers precise control over pore sizes, pore shapes and pore functionalities and constitutes a significant advancement in COF pore engineering.

## Results

### Synthesis and characterization

In this work, we employed a hexagonal-shaped Ph-An-COF as the parent network, which is a typical mesoporous COF with a concrete pore size of 2.9 nm[41]. Derived from Ph-An-COF, boronic ester-linked DBAAn-BTBA-COF was synthesized under similar solvothermal conditions through the condensation reaction using 4,4′-(2,3,6,7-tetrahydroxyanthracene-9,10-diyl)dibenzaldehyde (DBAAn) as linkers and 1,3,5-benzene-triboronic acid (BTBA) as vertices (Fig. 2a, Supplementary Figs. 1–4). The obtained precipitate was thoroughly washed with anhydrous acetone in a Soxhlet extractor and then dried under vacuum, affording DBAAn-BTBA-COF in an isolated yield of 89%. Its hexagonal skeleton is identical to that of Ph-An-COF, with benzene as corners but aldehyde-appended anthracene as walls. Fourier-transform infrared (FT-IR) spectroscopy supplied direct evidence for the presence of aldehyde groups with a typical vibration band at 1698 cm[−1], which was absent in the case of Ph-An-COF (Supplementary Fig. 5). This result indicates that the aldehyde units are well retained during the synthesis process of DBAAn-BTBA-COF. In the meantime, DBAAn-BTBA-COF exhibits a series

of vibration bands resulting from boronate ester linkages, which constitute the connection between linkers and corner blocks at 1242, 1334, and 1387 cm[−1]; these bands are the same as those observed in Ph-An-COF (Supplementary Fig. 5). Elemental analysis reveals the carbon and oxygen contents in DBAAn-BTBA-COF are close to its theoretical values for an infinite 2D skeleton (Supplementary Table 1). Solid-state cross-polarization/magic angle spinning carbon-13 nuclear magnetic resonance (CP/MAS [13]C NMR) analysis shows a series of signals at 198.6, 147.5, 131.5, 126.8, 123.6, 114.7, and 108.3 ppm (Supplementary Fig. 6), which can be attributed to the carbon atoms in DBAAn-BTBA-COF.

Notably, the linker DBAAn which consists of boronic acid and aldehyde groups allows covalent connection with diols and amine units into polygon networks, respectively. As depicted in Fig. 2a, six aldehyde groups were oriented to the center of 1D hexagonal channels on each layer in DBAAn-BTBA-COF. Such geometric features make it possible to fit in planar hexagonal amine-based building blocks with $C_6$ symmetry, which can further react with all six aldehydes simultaneously, leading to the formation of imine bonds and thereby partitioning one pore into six equal domains. According to the principle of symmetry and dimension matching, hexaminophenyl benzene (HAPB) was selected as the optimal baffle for the implementation of pore partition. The post-synthetic treatment was conducted in dioxane/mesitylene at 120 °C for 7 days, followed by washing with anhydrous acetone, affording DBAAn-BTBA-HAPB-COF in a yield of 84%. In the FT-IR spectrum of DBAAn-BTBA-HAPB-COF, a new stretching vibration band of C = N bond at 1622 cm[−1] appeared, while the vibration band of C = O at 1698 cm[−1] existing in that of DBAAn-BTBA-COF disappeared, indicating the complete transformation from C = O to C = N bonds (Supplementary Fig. 5). Meanwhile, the elemental analysis showed obvious composition changes, indicating successful integration of HAPB into the fresh network (Supplementary Table 1). In the CP/MAS [13]C NMR spectrum of DBAAn-BTBA-HAPB-COF, a set of peaks ranging from 106.7 to 153.4 ppm were clearly observed (Supplementary Fig. 7). Several new peaks at 151.3, 149.2, 115.8, and 106.7 ppm, which are absent in the spectrum of DBAAn-BTBA-COF, can be assigned to characteristic signals of hexaphenylbenzene. In addition, owing to the successful integration of the HAPB unit, a significant difference was observable in the diffuse reactance UV–visible absorption spectra of DBAAn-BTBA-COF and DBAAn-BTBA-HAPB-COF (Supplementary Fig. 8). Specifically, DBAAn-BTBA-COF showed a maximum adsorption peak at 396 nm, while DBAAn-BTBA-HAPB-COF exhibited a maximum adsorption peak at 437 nm.

Inspired by the double-stage strategy reported by Zeng et al. and Chen et al.[42,43], we attempted to synthesize DBAAn-BTBA-HAPB-COF through a one-pot procedure (Fig. 2b), in which linear bifunctional building blocks with boronic acid and aldehyde functionalities were employed for the construction of COFs with two types of covalent bonds. However, the obtained fluffy product was porous but almost amorphous (Supplementary Figs. 9 and 10), illustrating the one-pot strategy was not feasible for the construction of such a complex network. Moreover, we also tried to synthesize the imine-linked DBAAn-HAPB-COF as a parent network to realize pore partitioning (Supplementary Fig. 11), while it was obtained as an amorphous and nonporous powder (Supplementary Figs. 12 and 13). Even though the amorphous DBAAn-HAPB-COF was delivered to the subsequent reaction with $C_3$-symmetry BTBA (Supplementary Fig. 11), the resulting DBAAn-BTBA-HAPB-COF was also proved amorphous and nonporous (Supplementary Figs. 14 and 15). These results suggested that the highly crystalline parent framework might serve as a host scaffold to interlock with the baffle unit, which was crucial for the implementation of the pore partitioning strategy.

The crystalline structures of DBAAn-BTBA-COF and DBAAn-BTBA-HAPB-COF were resolved by powder X-ray diffraction (PXRD) measurement combined with the theoretical calculation. DBAAn-BTBA-COF exhibited a series of diffraction peaks at 3.42, 5.87, 6.78, 10.18, and 25.82, which were assigned to (100), (110), (200), (210), and (001)

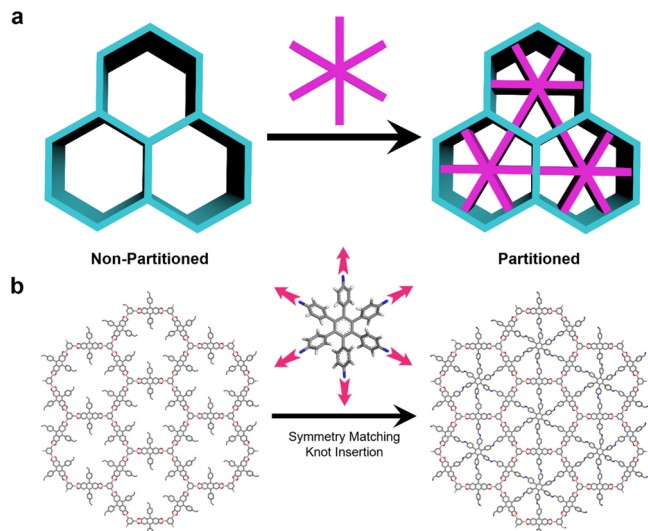

**Fig. 1 | Illustration of pore partition strategy. a** Graphical representation of pore partition in hexagonal channels. **b** Transformation of DBAAn-BTBA-COF into DBAAn-BTBA-HAPB-COF through symmetry-matching knot insertion.

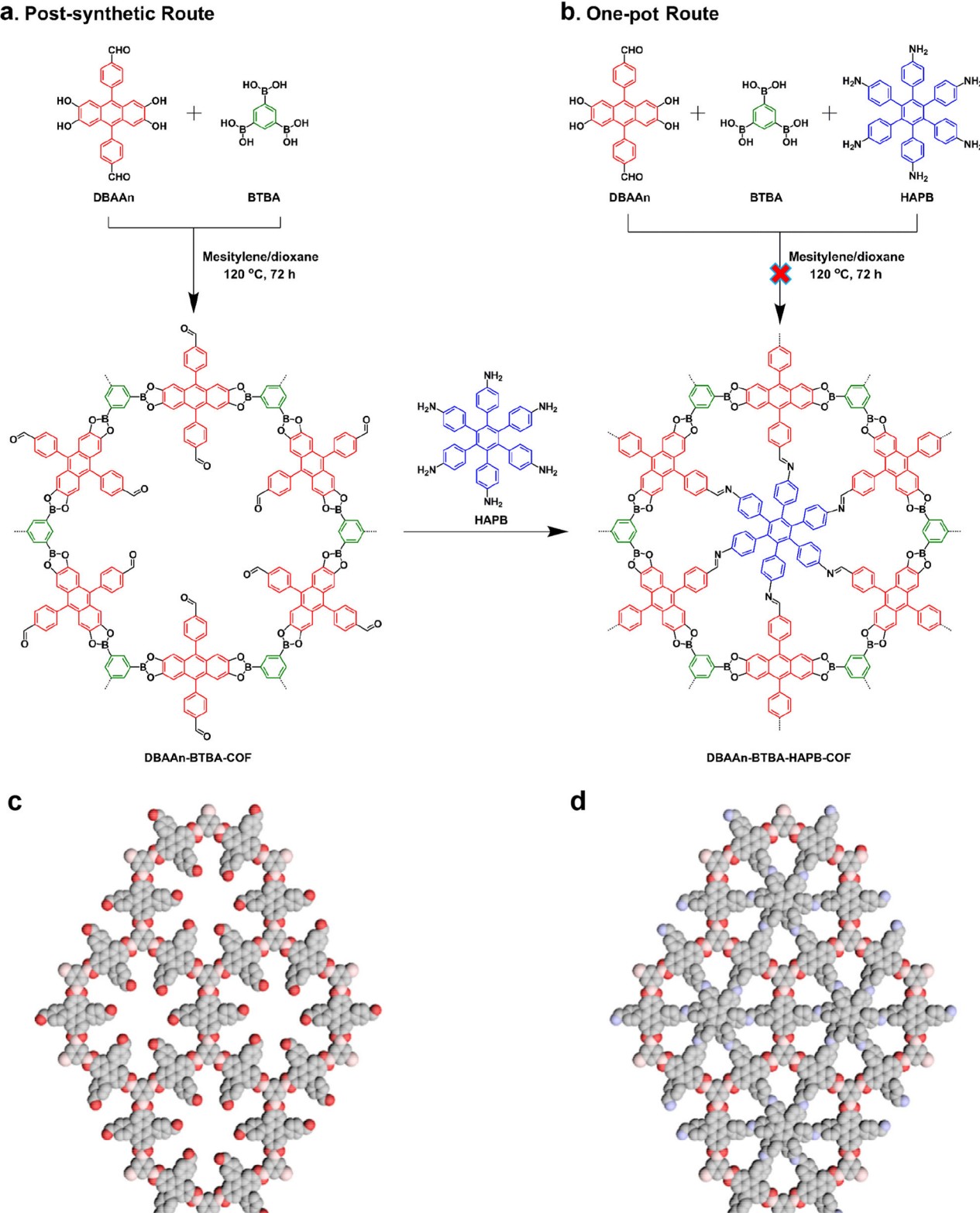

**Fig. 2 | Chemical structures of COFs. a** Post-synthetic route of DBAAn-BTBA-COF and DBAAn-BTBA-HAPB-COF. **b** One-pot synthetic route of DBAAn-BTBA-COF.
**c** Schematic diagram of DBAAn-BTBA-COF. **d** Schematic diagram of DBAAn-BTBA-HAPB-COF.

facets, respectively (Supplementary Fig. 16). By contrast, DBAAn-BTBA-HAPB-COF showed a set of PXRD signals at 3.38, 5.86, 6.76, 8.96, 10.16, 11.74, 12.22, 14.78, and 20.98, which were attributed to (100), (110), (200), (300), (210), (400), (500), (600), and (001) facets, respectively (Supplementary Fig. 17). The significant deviations in

diffraction peaks indicates the crystal transition from DBAAn-BTBA-COF to DBAAn-BTBA-HAPB-COF. Pawley refinement duplicates two sets of PXRD patterns, which are in good agreement with experimentally observed patterns, as evident by their negligible difference (Supplementary Figs. 16 and 17). Compared with staggered AB stacking

modes, the eclipsed AA stacking can reproduce their peak positions and relative intensity of these two COFs (Supplementary Figs. 18 and 19). A hexagonal unit cell (*P6*) with the parameters of a = 30.2358 Å, b = 30.2358 Å, c = 4.4387 Å, α = β = 90°, and γ = 120° was deduced for DBAAn-BTBA-COF (Supplementary Tables 2 and 3). In the skeleton of DBAAn-BTBA-COF, two kinds of center-to-center pore diameters exist in the 2D plane, one is 3.0 nm between the centers of opposite DBAAn linkers, the other one is 1.0 nm between corresponding two aldehydes (Fig. 2c). In comparison, hexagonal unit cell (P6) with parameters of a = 30.1380 Å, b = 30.1380 Å, c = 4.2305 Å, α = β = 90°, and γ = 120° was afforded for DBAAn-BTBA-HAPB-COF (Supplementary Tables 4 and 5). The calculated pore size of DBAAn-BTBA-HAPB-COF was determined as 6.5 Å (Fig. 2d), which was the smallest recorded value among all the reported COFs[2,26].

Field emission scanning electron microscopy (FE-SEM) shows that Ph-DBAAn-COF and Ph-DBAAn-HAPB-COF adopt micrometer-scale spherical and flocculent morphologies, respectively (Supplementary Figs. 20 and 21). High-resolution transmission electron microscopy (HR-TEM) images exhibited distinct porous textures (Supplementary Figs. 22 and 23). Thermal gravimetric analysis (TGA) reveals that both COFs were highly stable up to 450 °C (Supplementary Fig. 24). Nitrogen sorption isotherms were collected at 77 K to evaluate the porosity of DBAAn-BTBA-COF and DBAAn-BTBA-HAPB-COF. Notably, type IV and I isotherms were observed for DBAAn-BTBA-COF and DBAAn-BTBA-HAPB-COF, respectively (Supplementary Figs. 25 and 26). Brunauer-Emmett-Teller (BET) surface areas were calculated as 1056 and 348 m$^2$ g$^{-1}$ for DBAAn-BTBA-COF and DBAAn-BTBA-HAPB-COF, respectively. Owing to the occupation of aldehyde substituents and HAPB units in the 1D channels, they exhibited much lower porosity than that of Ph-An-COF[41]. The pore sizes of DBAAn-BTBA-COF and DBAAn-BTBA-HAPB-COF were respectively determined as 2.6 nm and 0.65 nm (Supplementary Figs. 27 and 28), which were highly consistent with their theoretical values. Accordingly, their pore volume values were determined as 0.62 nm and 0.35 cm$^3$ g$^{-1}$ (Supplementary Figs. 27 and 28), respectively.

## Hexane isomers vapor adsorption and separation

As is well known, the efficient separation of alkane isomers remains as an enormous challenge due to their chemical inertness and similar polarizabilities, leaving contour as the primary handgrip to distinguish them[44]. The separation is particularly significant for the production technology of petroleum, which contains multiple isomers of pentane and hexane. For example, hexanes in formula $C_6H_{14}$ contain five different isomers, accounting for 10–30% of the mixture. The five isomers differ greatly in their research octane number (RON), which constitutes an indicator to evaluate the anti-knock combustion ability of petroleum. The higher RON corresponds to the better quality of petroleum[45]. The RON values of *n*-hexane (*n*HEX), 2-methylpentane (2MP), 3-methylpentane (3MP), 2,3-dimethylbutane (23DMB), and 2,2-dimethylbutane (22DMB) are 30, 74, 75, 105, and 94, respectively. To obtain blends with higher RON values, zeolites are widely utilized as a sieve to separate these isomers, affording a mixture of four hexane isomers with an average RON of 83. Sieving effect has been widely acknowledged as a powerful tool to realize the separation and purification of mixtures through regulating shapes and dimensions of pores that allow smaller molecules to go through and prevent larger ones[46,47]. For example, various metal-organic frameworks (MOFs), such as Fe$_2$(BDP)$_3$[48], Zr-bpdc[49], and MIL-53(Fe)-(CF$_3$)$_2$[50], have been developed and showed excellent performance in the separation of alkane isomers. Similar to these MOFs, DBAAn-BTBA-HAPB-COF has ultra-microporous channels and wedge-shaped corners, making it also quite perspective in this respect.

Equilibrium adsorption isotherms of these hexane isomers were tested at 100, 120, and 150 °C (Fig. 3), all of which fall in the range of industrial separation temperature. It's noticeable that DBAAn-BTBA-HAPB-COF shows varied adsorption capability toward different hexane isomers. At 100 °C, the saturated adsorption capacities for *n*HEX, 2MP, 3MP, 23DMB, and 22DMB were determined as 1.22, 1.05, 1.02, 0.98, and 0.97 mmol g$^{-1}$, respectively (Fig. 3a–e). These adsorption isotherms rise with different slopes until reaching saturation. The descending slopes in their isotherms for linear versus monobranched versus dibranched isomers demonstrate a homologous decline in the adsorption strength. A similar tendency was also observed for the adsorption of hexane isomers by Fe$_2$(BDP)$_3$, Zr-bptc, and RE-fcu-MOF[48,49,51]. Correspondingly, the saturation capacities were 0.99, 0.81, 0.78, 0.76, and 0.75 mmol g$^{-1}$ at 120 °C, and 0.58, 0.34, 0.32, 0.24, and 0.25 mmol g$^{-1}$ at 150 °C for *n*HEX, 2MP, 3MP, 23DMB, and 22DMB, respectively (Fig. 3a–e). Obviously, saturated adsorption amounts were attained or almost attained at 100 and 120 °C, whereas higher pressure was required at 150 °C. Significantly, these high adsorption capacities of DBAAn-BTBA-HAPB-COF toward five hexane isomers were comparable with several landmark adsorbents, such as Fe$_2$(BDP)$_3$, MIL-101(Cr), and zeolite 5A (Supplementary Table 6)[48,52,53]. In addition, the isosteric heats of adsorption ($Q_{st}$) were determined according to their isotherms obtained at 100, 120, and 150 °C (Supplementary Fig. 29). Notably, *n*HEX showed the highest $Q_{st}$ value among the five isomers, illustrating the strongest interaction occurs between this isomer and DBAAn-BTBA-HAPB-COF. This is because a larger fraction of its surface can interact with the 1D channel pore surface of DBAAn-BTBA-HAPB-COF than the other isomers. In comparison, the monobranched 2MP and 3MP exhibited slightly lower $Q_{st}$ values owing to shorter chains interplaying with the pore surface. The dibranched 23DMB and 22DMB are too compact and not flexible enough to create strong van der Waals interaction with the channel surfaces, leading to the lowest $Q_{st}$ values. This phenomenon was also observed at low loadings in other porous materials[48,53] due to these isomers residing at different sites in a heterogeneous porous structure. Therefore, it can be inferred that the adsorption heats of these isomers have a direct correlation with their degree of branching.

To investigate the separation efficiency of DBAAn-BTBA-HAPB-COF for hexane isomers, a breakthrough experiment was conducted to simulate real-world working conditions. An equimolar quinary mixture of hexane isomers in helium gets through a packed bed with DBAAn-BTBA-HAPB-COF as filling porous agent at a flow velocity of 2.5 cm$^3$ min$^{-1}$. At 100 °C, an eluting sequence of five isomers was achieved as follows: 23DMB, 22DMB, 3MP, 2MP, and *n*HEX, whose respective retention time increased in the same order (Supplementary Fig. 30). This result indicated that DBAAn-BTBA-HAPB-COF could work as an efficient sieve to realize complete separation of these hexane isomers. In addition, the RON value of the mixed products coming out from the packed column was also plotted, simplified as a weighted average of the RONs of each component. It is significant to note that the RON value of the eluted mixture was on the verge of 95 in the preliminary stage of the breakthrough experiment and then gradually decayed to 78. When the temperature rises to 120 °C, the most desirable dibranched isomers, 22DMB first eluted from the column, subsequently followed by 23DMB, monobranched 2MP, 3MP, and finally *n*HEX (Fig. 3f). In the meanwhile, the initial average RON value for these isomers was calculated as 99, which was much higher than that of industrial grade mixture of hexane isomers. Such an elution order is consistent with those of other porous materials, such as Fe$_2$(BDP)$_3$, Zr-bptc, and RE-fcu-MOF[48,49,51]. Notably, linear *n*HEX was eluted first from the packed column, immediately followed by the mixture of the other four isomers at 150 °C (Supplementary Fig. 31), indicating the operation temperature exerts an enormous influence on the elution sequence of these hexane isomers. The elution sequence can be attributed to the diffusion rate of these isomers, which is related to the separation temperature and pore aperture. Furthermore, the sieving effect is another critical factor to be considered that exerts significant influence over the elution

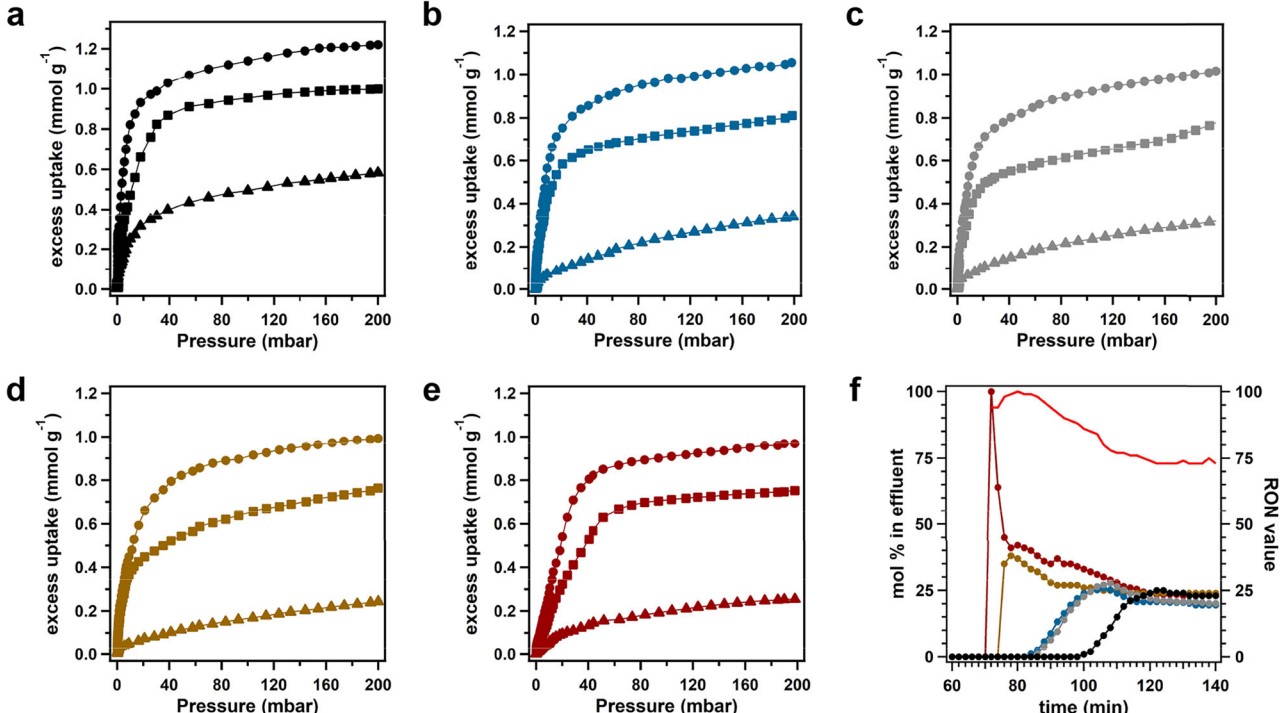

**Fig. 3 | Pure-component equilibrium adsorption isotherms and breakthrough results. a–e** Gas adsorption isotherms for *n*HEX (black), 2MP (blue), 3MP (gray), 23DMB (brown), and 22DMB (maroon) in DBAAn-BTBA-HAPB-COF at 100 °C (dots), 120 °C (squares), and 150 °C (triangles). **f** Breakthrough curves of an equimolar mixture of 22DMB (maroon line), 23DMB (brown line), 2MP (blue line), 3MP (gray line), and *n*HEX (black line) in nitrogen at 1 atm, and these data sets correspond to the left *Y*-axis. The red line is the calculated RON values for eluted mixture, which corresponds to the right *Y*-axis.

sequence needs. As a consequence, a synergetic effect of sieving effect and diffusion was proposed in the elution dynamics for the separation of these hexane isomers. As control experiments, the breakthrough tests using Ph-An-COF and Ph-DBAAn-COF as porous packing was further evaluated at 120 °C. However, both of them showed almost the same separation performance for the five hexane isomers (Supplementary Figs. 32 and 33), which can be attributed to their inappropriate pore shapes and dimensions.

### Theoretical calculations

We further conducted configurational-bias Monte Carlo (CBMC) simulations to clarify the mechanism of separation of hexane isomers in the breakthrough experiment. As shown in Fig. 4a–c, Ph-An-COF, DBAAn-BTBA-COF, and DBAAn-BTBA-HAPB-COF exhibit hexagonal, hexagram, and wedgy channels, respectively. The maximum pore sizes associated with van der Waals surfaces are 29, 26, and 6.5 Å for Ph-An-COF, DBAAn-BTBA-COF, and DBAAn-BTBA-HAPB-COF, respectively (Fig. 4d–i). The kinetic diameters of *n*HEX, 2MP, 3MP, 22DMB, and 23DMB are calculated as 4.3, 5.0, 5.0, 5.6, and 6.2 Å, respectively (Fig. 4j). Obviously, the pore size of DBAAn-BTBA-HAPB-COF is closest to the kinetic diameters (Φ) of the five hexane isomers. Therefore, the well-aligned ultramicroporous channels are supposed to serve as efficient sieves for the separation of the five isomers of hexane (Fig. 5a, b). In addition, the calculation results are revealed as snapshots of the conformations and locations of the five isomers adsorbed within the 1D wedgy channels of DBAAn-BTBA-HAPB-COF (Fig. 5c). The number of each isomer differs a lot in each snapshot of arbitrary channel segments, in spite of the same loading of these molecules in each unit cell before computational simulation. Notably, the van der Waals overlap between each isomer and pore wall of DBAAn-BTBA-HAPB-COF decreases with their degrees of branching (Supplementary Figs. 34–36). A similar trend was also identified in the diffusion process of alkane isomers in

Fe₂(BDP)₃ and zeolite NaY[48,54]. The backbones of hexane isomers line up along the centered vertices of the 1D wedgy channels, thus making the largest pore space for dispersion interactions. Furthermore, it's quite distinct that the number of carbon atoms effectively interacting with the pore wall of DBAAn-BTBA-HAPB-COF declines with the degree of branching, which was verified by the observed conformations of these hexane isomers. The linear isomer is the most flexible, while the dimethylbutane isomers are the most compact. The order of van der Waals interactions with the framework surface is as follows: *n*HEX > 3MP > 2MP > 23DMB > 22DMB, which is just opposite to their eluting sequence.

### Discussion

In summary, we have developed a pore partition strategy in 2D COFs through the insertion of a third symmetry-matching regulated building block into a prebuilt skeleton. Two types of covalent linkages were introduced within one framework via stepwise pathways. Precise control over pore properties, including pore shape, pore dimension, and pore environment, can be implemented well. The obtained ultramicroporous COFs with wedge-shaped corners are potentially employed as highly efficient sieves for the separation of hexane isomers. Pure gas sorption test reveals that DBAAn-BTBA-HAPB-COF exhibits selective adsorption capacities toward the five hexane isomers, owing to their different interactions with pore surfaces. Breakthrough experiment and computational simulation further verified that DBAAn-BTBA-HAPB-COF could serve as a high-efficiency filling material to realize the separating of branched hexanes based on sieving effects, affording blends of C₆ alkane isomers with high RON values, which are comparable with that of Fe₂(BDP)₃ and zeolite 5A. More significantly, this work not only provides an interesting insight into the pore partition of mesoporous 2D COFs by tuning the geometry and reactive sites of primary and secondary building blocks but also opens up numerous opportunities to regulate and control the

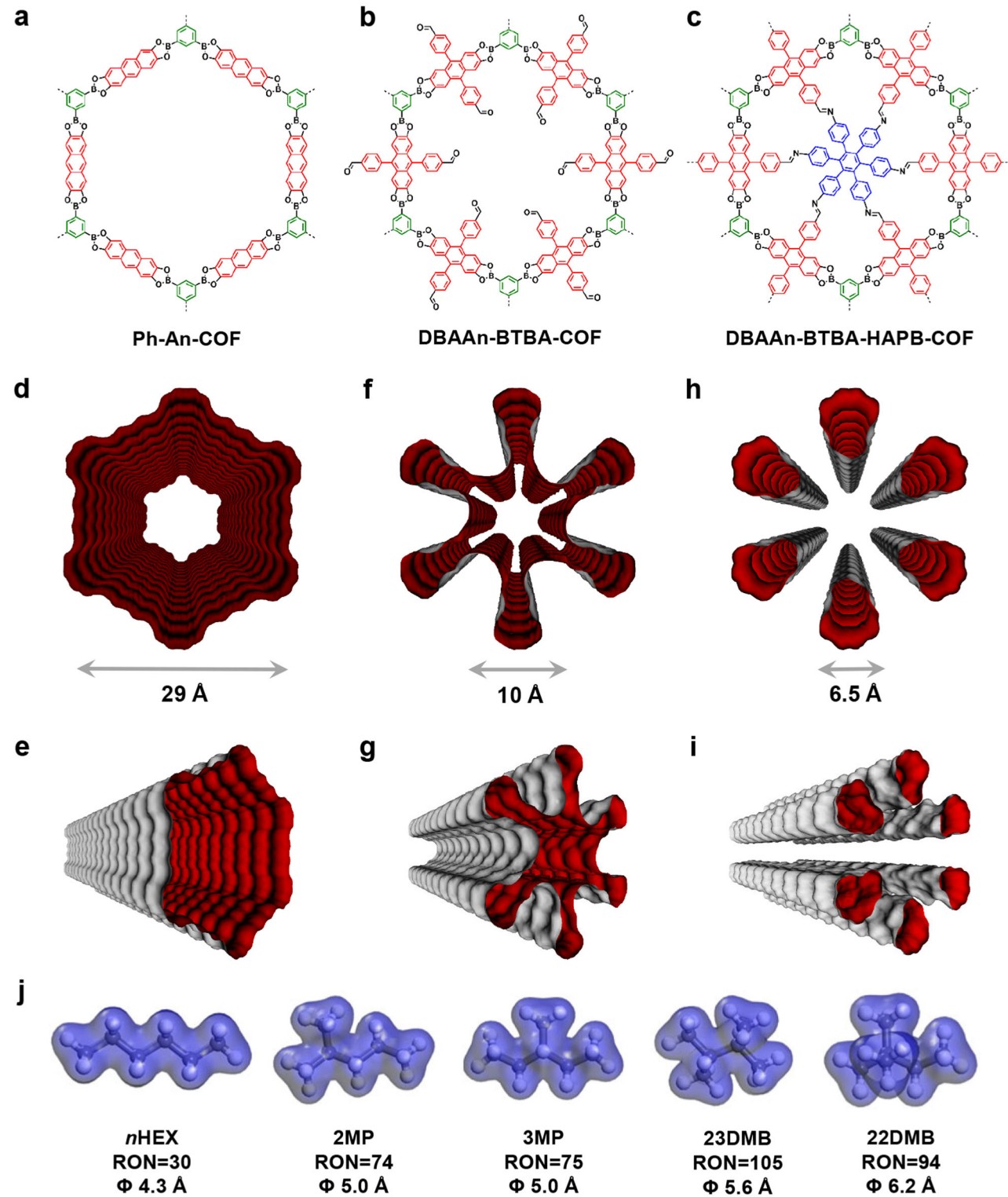

**Fig. 4 | Structures and van der Waals surfaces of COFs. a–c** Structural formula of Ph-An-COF, DBAAn-BTBA-COF, and DBAAn-BTBA-HAPB-COF. **d, e** Top-view and side-view van der Waals surfaces associated with the 1D channels running through Ph-An-COF. **f, g** Top-view and side-view van der Waals surfaces associated with the 1D channels running through DBAAn-BTBA-COF. **h, i** Top-view and side-view van der Waals surfaces associated with the 1D channels running through DBAAn-BTBA-HAPB-COF. **j** Hexane isomers (*n*HEX, 2MP, 3MP, 23DMB, and 22DMB) with their RONs and Φ values.

properties and functionalities of COFs. Therefore, the pore partition might work as a promising tool to implement the construction of microporous and ultramicroporous COFs, greatly expand the diversity and complexity of this material, and actively push forward the process in various practical applications.

## Methods
### Synthesis of DBAAn-BTBA-COF
A dioxane/mesitylene (5/5 by vol.; 1 mL) mixture of 1,3,5-benzenetriboronic acid (0.024 mmol, 5.00 mg) and 2,3,6,7-tetrahydroxyanthracene (0.036 mmol, 16.12 mg) in a Pyrex tube (10 mL)

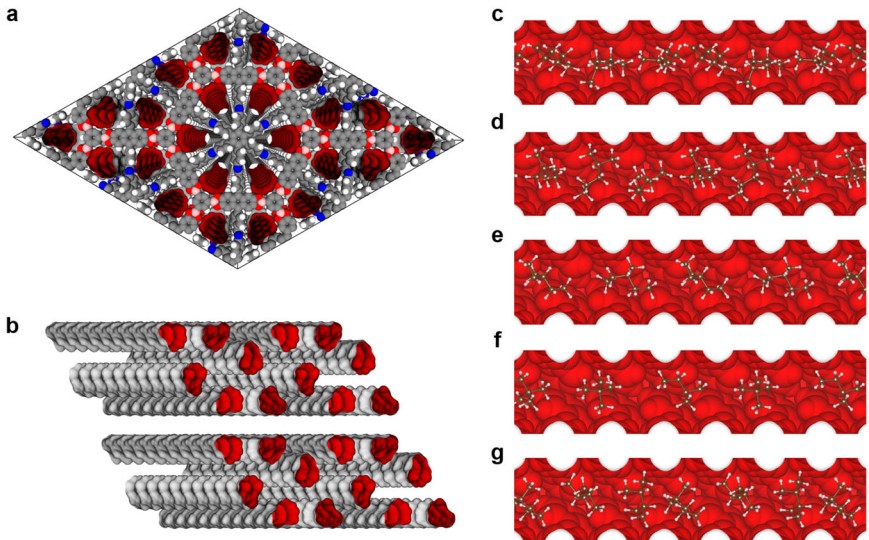

**Fig. 5 | Theoretical calculation. a** Schematic diagram of DBAAn-BTBA-HAPB-COF and its calculational van der Waals surfaces per unit cell. **b** Calculational van der Waals channels of DBAAn-BTBA-HAPB-COF per unit cell. **c**–**g** Snapshots of *n*HEX, 2MP, 3MP, 23DMB, and 22DMB within van der Waals channels of DBAAn-BTBA-HAPB-COF at 120 °C.

was degassed by three freeze-pump-thaw cycles. After that, the tube was sealed off and heated to 120 °C for 3 days. The precipitates were collected by centrifugation at a speed of 2800 × *g* for 5 min and then washed with anhydrous tetrahydrofuran (3 × 10 mL) and acetone (3 × 10 mL) until the eluate was colorless. The powder was dried at 80 °C under vacuum overnight to give DBAAn-BTBA-COF as a dark-yellow powder with a yield of 89%. In this method, the maximum synthesis quantity of DBAAn-BTBA-COF can be scaled up to about 550 mg without significant loss of its crystallinity and porosity.

### Synthesis of DBAAn-BTBA-HAPB-COF
**Post-synthetic route.** A mixture of DBAAn-BTBA-COF (18.65 mg) and hexaaminophenyl benzene (0.012 mmol, 7.45 mg) in a Pyrex tube (10 mL) with dioxane (1 mL) was degassed by three freeze-pump-thaw cycles. Then the tube was sealed off and heated to 120 °C for 7 days. The precipitates were collected by centrifugation at a speed of 2800 × *g* for 5 min and then washed with anhydrous acetone (3 × 10 mL). The powder was dried at 80 °C under vacuum overnight, affording DBAAn-BTBA-HAPB-COF as a greenish-yellow powder in a yield of 84%. In this method, the maximum synthesis quantity of DBAAn-BTBA-HAPB-COF can be scaled up to about 280 mg without significant loss of its crystallinity and porosity.

**One-pot route.** A mixture of 1,3,5-benzenetriboronic acid (0.024 mmol, 5.00 mg), 2,3,6,7-tetrahydroxyanthracene (0.036 mmol, 16.12 mg), and hexaaminophenyl benzene (0.012 mmol, 7.45 mg) in a Pyrex tube (10 mL) with dioxane/mesitylene (5/5 by vol.; 1 mL) was degassed by three freeze-pump-thaw cycles. Then the tube was sealed off and heated to 120 °C for 3 days. The precipitates were collected by centrifugation at a speed of 2800 × *g* for 5 min and then washed with anhydrous tetrahydrofuran (3 × 10 mL) and acetone (3 × 10 mL) until the eluate was colorless. The powder was dried at 80 °C under vacuum overnight, affording DBAAn-BTBA-HAPB-COF as a greenish-yellow powder in a yield of 83%. In this method, the maximum synthesis quantity of DBAAn-BTBA-HAPB-COF can be scaled up to about 320 mg without significant loss of its crystallinity and porosity.

### Gas adsorption measurements
For all gas adsorption measurements, COFs (100 mg) were activated under vacuum at 100 °C for 24 h. The COFs samples were then transferred to Micromeritics ASAP 2020 gas adsorption analyzer. Nitrogen adsorption measurements were conducted using 99.999% purity gas. All the hexane isomers were purchased from TCI and were added to the vapor adsorption apparatus of ASAP 2020. The hexane isomers were frozen by liquid nitrogen. The pressure was monitored before reaching 0.001 mbar, and then the headspace was evacuated for 5 min. After three freeze-pump-thaw cycles, the liquids were distilled into a new tube three times, each time only collecting approximately 50% of the original volume.

### Breakthrough measurement
COF powder was packed into a stainless-steel column with an inner diameter of 4.0 mm and a length of 150 mm. A helium flow was used to purge the adsorbent at a rate of 2.5 mL/min. For a typical measurement, pure helium was bubbled into a mixture of hexane isomers at a rate of 2.5 mL/min. The volumes of 2,2-dimethylbutane, 2,3-dimethylbutane, 2-methylpentane, 3-methylpentane, and n-hexane were 2.67, 3.50, 3.79, 4.22, and 5.82 mL, respectively. All column breakthrough experiments were performed at 298 K and 100 kPa. The composition of all gases was monitored using an Agilent 6890N gas chromatograph with a flame ionization detector.

### Computational calculation
The crystalline structures of COFs were determined using the density-functional tight-binding (DFTB) method. The calculations were conducted using the DFTB⁺ program package version 17.1. Lennard-Jones-type dispersion was utilized in these calculations to determine van der Waals and possible π-stacking interactions. Metropolis Monte Carlo method was employed for possible adsorption configuration searching, and the COMPASS II force field was utilized for calculating charges and non-bond interactions. The geometries of obtained adsorption models were optimized using Gaussian 16 Rev. C.01 package at M06-2X exchange-correlation functional in combination with cc-pvtz basis set. For the optimized adsorption models, cc-pvtz was selected as the basis set combined with M06-2X as the exchange-correlation functional. The adsorption energies were determined for the basis set superposition error (BSSE) with the counterpoise method.

### Data availability
All data supporting the findings of this study are available within the article, as well as the Supplementary Information file. All other data

supporting the findings of the study are available from the corresponding author upon request.

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

## Acknowledgements

This work is supported by financial support from the National Natural Science Foundation of China (92163131), the National Key Research and Development Program of China (2022YFE0130700), and the Fundamental Research Funds for the Central Universities (226-2023-00113), all of which were awarded to N.H.

## Author contributions

N.H. conceived the research and designed experiments. X.X. synthesized all the compounds and COFs. X.X., X.W., and K.X. contributed to the material characterization and data analysis. H.X. conducted computational simulations of related materials. X.X., H.C., and N.H. co-wrote the manuscript, and all authors discussed the results and commented on the manuscript.

## Competing interests
The authors declare no competing interests.
