## [Peer Review File · Nature Communications]

Pore partition in two-dimensional covalent organic frameworksReviewers' Comments:

Reviewer #1:

Remarks to the Author:

The manuscript describes a neat synthetic trick to partition space within a covalent organic framework (COF). The structural data is typical for most COF studies in that it shows very broad powder XRD patterns characteristic will only local order and rather unconvincing modelling to elucidate the structure. However, such analysis is very prevalent in the COF field and so it would be unfair to single out this study for doing what everybody else does! That said, the basic concept demonstrated of partitioning small is confirmed by nice adsorption data. I am sure that other researchers on COFs will find this interesting. Therefore, I am happy to recommend acceptance if the authors can add the following to the manuscript.

The experimental describes a synthesis on a 15 mg scale and yet the adsorption studies must have been carried out on >100 mg of sample. I am curious if the synthesis can be scaled up (or were multiple batches made on a 15 mg scale?). Some comments on scalability should be made.

The standard of technical English falls short for publication and the authors should improve on this for the next submission.

Reviewer #2:

Remarks to the Author:

In this work, the authors used a pore partition strategy for segmentation of a mesopore into multiple ultramicroporous domains in covalent organic frameworks (COFs), where the pore partition is achieved by inserting a rigid building block into a prebuilt parent framework. The resulted COFs were employed for the separation of hexane isomers based on sieving effect.

My major concern is whether the authors really obtained such a delicate system (DBAAn-BTBA-HAPB-COF) from DBAAn-BTBA-COF and HAPB through the pore partition, since there is no direct experimental evidence to demonstrate that they indeed prepared DBAAn-BTBA-HAPB-COF. How can the authors make sure that all the six amino groups of each HAPB could perfectly react with six aldehyde groups of each pore? Are there structural defects?

To further support the possibility for the formation of DBAAn-BTBA-HAPB-COF, the authors may synthesize a spoked wheel-like fragment as a model compound, similar to the report by Anderson et al (Acc. Chem. Res. 2018, 51, 2083–2092).

Reviewer #1:

The manuscript describes a neat synthetic trick to partition space within a covalent organic framework (COF). The structural data is typical for most COF studies in that it shows very broad powder XRD patterns characteristic will only local order and rather unconvincing modelling to elucidate the structure. However, such analysis is very prevalent in the COF field and so it would be unfair to single out this study for doing what everybody else does! That said, the basic concept demonstrated of partitioning small is confirmed by nice adsorption data. I am sure that other researchers on COFs will find this interesting. Therefore, I am happy to recommend acceptance if the authors can add the following to the manuscript.

We appreciate the comments.

The experimental describes a synthesis on a 15 mg scale and yet the adsorption studies must have been carried out on >100 mg of sample. I am curious if the synthesis can be scaled up (or were multiple batches made on a 15 mg scale?). Some comments on scalability should be made.

We appreciate the comments. Actually, the synthesis of most COFs in sealed tubes can be scaled up to hundreds of milligrams without significant loss of crystallinity and porosity. In this work, the synthesis quantity of DBAAn-BTBA-COF and DBAAn-BTBA-HAPB-COF can be scaled up to about 550 and 320 mg, respectively. The corresponding comments have been added on Pages 13 and 14 in the revised manuscript.

The standard of technical English falls short for publication and the authors should improve on this for the next submission.

We appreciate the comments. We have polished the language of the revised manuscript.

Reviewer #2:

In this work, the authors used a pore partition strategy for segmentation of a mesopore into multiple ultramicroporous domains in covalent organic frameworks (COFs), where the pore partition is achieved by inserting a rigid building block into a prebuilt parent framework. The resulted COFs were employed for the separation of hexane isomers based on sieving effect.

We appreciate the comments.

My major concern is whether the authors really obtained such a delicate system (DBAAn-BTBA-HAPB-COF) from DBAAn-BTBA-COF and HAPB through the pore partition, since there is no direct experimental evidence to demonstrate that they indeed prepared DBAAn-BTBA-HAPB-COF. How can the authors make sure that all the six amino groups of each HAPB could perfectly react with six aldehyde groups of each pore? Are there structural defects?

Scheme R1. Synthetic route to spoked wheel-like molecule **6**.

We appreciate the comments.

To verify that all the six amino groups of each HAPB can perfectly react with six aldehyde groups of each pore, we synthesized a spoked wheel-like molecule **6** as a model compound (**Scheme R1**). Compound **6** can be obtained through the reaction between compound **5** and HAPB. This result presents the feasibility that DBAAn-BTBA-HAPB-COF can be obtained from DBAAn-BTBA-COF and HAPB through the pore partition strategy. Most reported COFs are obtained as powdered crystal rather than single crystal, some structural defects do exist in this kind of material (*Science* **2017**, *355*, 923; *Nat. Rev. Mater.* **2016**, *1*, 16068;

Chem. Rev. **2020**, *120*, 8814–8933; *Nat. Rev. Methods Primers* **2023**, *3*, 1). Therefore, there are also structural defects existing in the obtained DBAAn-BTBA-HAPB-COF.

To further support the possibility for the formation of DBAAn-BTBA-HAPB-COF, the authors may synthesize a spoked wheel-like fragment as a model compound, similar to the report by Anderson et al (*Acc. Chem. Res.* 2018, *51*, 2083–2092).

We appreciate the comments.

As suggested by the reviewer, we attempted to synthesize spoked wheel-like fragments **6** as model compound through five-step reactions (**Scheme R1**). However, the conjugated, rigid, and planar structure makes this compound very poorly soluble in various solvents. Compound **6** can be characterized using only MALDI-MS (**Figure R1**) and elementary analysis and (**Table R1**). Both results are consistent with their theoretical values. The synthetic routes and characterization results for compounds **1-6** were summarized in the following pages.

Synthesis routes of compounds 1-6

Synthesis of compound 1. In a 50 mL round bottom flask equipped with a magnetic stir bar, boron trifluoride diethyl etherate (2.71 mL, 10.00 mmol) was added dropwise to a solution of benzaldehyde (1.06 g, 10.00 mmol) and veratrole (2.76 g, 20.00 mmol) in dichloromethane (20 mL). After addition, the suspension was stirred for 8 h at room temperature. The reaction progress was monitored by TLC. After full conversion was achieved, the solvent was removed under reduced pressure. The crude product was purified by column chromatography on silica gel, eluting with petroleum ether/ dichloromethane. After evaporation, the resulting product was dried at 80 °C for 12 h to give compound **1** as white solid (2.84 g, 78%). ¹H NMR (500 MHz, CDCl₃) δ 7.29 (t, *J* = 7.5 Hz, 2H), 7.21 (d, *J* = 7.3 Hz, 1H), 7.11 (d, *J* = 7.3 Hz, 2H), 6.78 (d, *J* = 8.3 Hz, 2H), 6.67 (d, *J* = 1.9 Hz, 2H), 6.60 (dd, *J* = 8.3, 1.9 Hz, 2H), 5.44 (s, 1H), 3.86 (s, 6H), 3.76 (s, 6H). ¹³C NMR (101 MHz, Chloroform-*d*) δ 148.86, 147.57, 144.36, 136.79, 129.29, 128.25, 126.26, 121.47, 112.95, 111.02, 55.96, 55.89, 55.85.

Synthesis of compound 2. In a 50 ml round-bottom flask equipped with a magnetic stir bar, a solution of **1** (2.84 g, 7.79 mmol) and terephthalaldehyde (1.15 g, 8.57 mmol) in dichloromethane (5 mL) was added dropwise to 84% sulfuric acid (10 mL) at 0 °C. After addition, the suspension was stirred for 2 h at room temperature. The reaction progress was monitored by TLC. After full conversion was achieved, the reaction was quenched with water and neutralized by ammonia. After extraction with dichloromethane, the solvent was removed under reduced pressure. The crude product was purified by column chromatography on silica gel, eluting with dichloromethane. After evaporation, the resulting product was dried at 80 °C for 12 h to give compound **2** as light-yellow solid (0.48 g, 13%). ¹H NMR (500 MHz, CDCl₃) δ 10.21 (s, 1H), 8.15 (d, *J* = 7.9 Hz, 2H), 7.70 (d, *J* = 7.9 Hz, 2H), 7.62 (t, *J* = 7.4 Hz, 2H), 7.54 (t, *J* = 7.4 Hz, 1H), 7.48 (d, *J* = 7.2 Hz, 2H), 6.83 (s, 2H), 6.70 (s, 2H), 3.73 (d, *J* = 5.1 Hz, 12H). ¹³C NMR (101 MHz, CDCl₃) δ 192.03, 149.31, 148.97, 147.06, 139.58, 135.62, 133.92, 132.02, 130.93, 130.12, 128.74, 127.56, 127.43, 125.84, 125.43, 104.20, 103.29, 55.52, 55.50.

Synthesis of compound 3. In a 100 ml round-bottom flask equipped with a magnetic stir bar, a solution of **2** (0.48 g, 1 mmol) was dispersed in hydrobromic acid aqueous solution (10 mL). The reaction mixture was stirred at reflux overnight and a large amount of green solid formed. After filtration, the resulting product

was dried at 40 °C for 12 h to give compound **3** as yellow greenish solid (0.29 g, 68%). ¹H NMR (500 MHz, DMSO-*d*₆) δ 10.20 (s, 1H), 9.39 (s, 4H), 8.17 (d, *J* = 7.9 Hz, 2H), 7.63 (t, *J* = 6.7 Hz, 4H), 7.54 (t, *J* = 7.4 Hz, 1H), 7.37 (d, *J* = 7.2 Hz, 2H), 6.67 (s, 2H), 6.59 (s, 2H). ¹³C NMR (101 MHz, DMSO-*d*₆) δ 192.95, 147.05, 146.49, 146.19, 139.98, 135.16, 131.96, 131.03, 130.82, 129.81, 128.91, 128.64, 127.08, 125.04, 124.67, 106.61, 105.95.

Synthesis of compound 5. In a 20 ml Schlenk flask equipped with a magnetic stir bar, 1,3-benzenediboric acid (19.62 mg, 0.118 mmol) and compound **3** (50.00 mg, 0.118 mmol) was dispersed in dioxane (5 mL) and then degassed by three freeze-pump-thaw cycles. The reaction mixture was stirred at 120 °C for 5 days. The precipitates were collected by centrifugation and then washed with anhydrous tetrahydrofuran (3×10 mL) and a small amount of acetone. The crude product was further purified by Soxhlet extraction with toluene. After evaporation, the resulting product was dried at 80 °C for 12 h to give compound **5** as brown solid (7.33 mg, 13 %). ¹H NMR (400 MHz, DMSO-*d*₆) δ 10.20 (s, 6H), 8.16 (dd, *J* = 8.1, 3.8 Hz, 12H), 7.84 – 7.79 (m, 18H), 7.63 (dd, *J* = 7.8, 3.5 Hz, 30H), 7.39 – 7.36 (m, 12H), 7.28 (s, 6H), 7.15 (s, 6H), 6.68 (d, *J* = 7.4 Hz, 6H), 6.60 (d, *J* = 2.9 Hz, 6H), 6.44 (d, *J* = 9.1 Hz, 6H), 6.38 (d, *J* = 4.1 Hz, 6H). Due to the poor solubility, the ¹³C NMR spectrum of compound **5** is unrecognizable. HRMS (MALDI-TOF): *m/z* [*M*]⁺ calculated for C₁₉₈H₁₀₈B₁₂O₃₀ 3096.8042; found 3096.6031.

Synthesis of compound 6. A mixture of **5** (12.38 mg, 0.004 mmol) and hexaaminophenyl benzene (2.50 mg, 0.004 mmol) in a Pyrex tube (10 mL) with dioxane (1 mL) was degassed by three freeze pump-thaw cycles. Then the tube was sealed off and heated to 120°C for 7 days. The precipitates were collected by centrifugation and then washed with anhydrous acetone (3×10 mL). The powder was dried at 80 °C under vacuum overnight to give compound **6** as greenish yellow solid (5.99 mg, 38%). Due to the poor solubility, ¹H and ¹³C NMR spectra of compound **6** are not available. HRMS (MALDI-TOF): *m/z* [*M*]⁺ calculated for C₂₄₀H₁₃₂B₁₂N₆O₂₄ 3613.0410; found 3613.9825.

Besides the above-mentioned compounds, we also synthesized several other building blocks (compounds **3b**, **3c**, and **4b**) to increase the solubility of the target model compound (as shown below). However, it's not as we expected, the macrocyclic compounds **5b** and **5c** exhibit quite poor solubility in most solvents, and are difficult to be characterized. Therefore, the further synthesis of target compounds **6b** and **6c** also failed.

Characterization of compounds 1-6

Figure R1. ^1H NMR spectrum of compound 1.

Figure R2. ^{13}C NMR spectrum of compound 1.

Figure R3. ^1H NMR spectrum of compound 2.

Figure R4. ^{13}C NMR spectrum of compound 2.

Figure R5. ¹H NMR spectrum of compound 3.

Figure R6. ¹³C NMR spectrum of compound 3.

Figure R7. ¹H NMR spectrum of compound 5.

Figure R8. ¹³C NMR spectrum of compound 5.

Figure R9. MALDI-MS spectra of compound 5.

Figure R10. MALDI-MS spectra of compound 6.

Table R1. Elemental analysis of compounds **5** and **6**.

Compound		C%	H%	N%
5	Calculated	76.80	3.52	---
	Found	76.78	3.53	---
6	Calculated	79.78	3.68	2.33
	Found	79.73	3.71	2.34

Reviewers' Comments:

Reviewer #1:

Remarks to the Author:

...I am happy for this paper to be accepted.

Reviewer #2:

Remarks to the Author:

The revised manuscript is now recommended for publication.

Point-by-point response to reviewers

Reviewer #1 (Remarks to the Author):

...I am happy for this paper to be accepted.

We appreciate the comments.

Reviewer #2 (Remarks to the Author):

The revised manuscript is now recommended for publication.

We appreciate the comments.